# Modified Protein-Water Interactions in CHARMM36m for Thermodynamics and Kinetics of Proteins in Dilute and Crowded Solutions

**DOI:** 10.3390/molecules27175726

**Published:** 2022-09-05

**Authors:** Daiki Matsubara, Kento Kasahara, Hisham M. Dokainish, Hiraku Oshima, Yuji Sugita

**Affiliations:** 1Laboratory for Biomolecular Function Simulation, RIKEN Center for Biosystems Dynamics Research, Kobe 650-0047, Hyogo, Japan; 2Division of Chemical Engineering, Graduate School of Engineering Science, Osaka University, Toyonaka 560-8531, Osaka, Japan; 3Theoretical Molecular Science Laboratory, RIKEN Cluster for Pioneering Research, Wako 351-0198, Saitama, Japan; 4Computational Biophysics Research Team, RIKEN Center for Computational Science, Kobe 650-0047, Hyogo, Japan

**Keywords:** molecular dynamics simulation, enhanced sampling method, molecular force fields, van der Waals interaction, CHARMM36m, NBFIX, intrinsically disordered proteins, crowding simulations

## Abstract

Proper balance between protein-protein and protein-water interactions is vital for atomistic molecular dynamics (MD) simulations of globular proteins as well as intrinsically disordered proteins (IDPs). The overestimation of protein-protein interactions tends to make IDPs more compact than those in experiments. Likewise, multiple proteins in crowded solutions are aggregated with each other too strongly. To optimize the balance, Lennard-Jones (LJ) interactions between protein and water are often increased about 10% (with a scaling parameter, λ = 1.1) from the existing force fields. Here, we explore the optimal scaling parameter of protein-water LJ interactions for CHARMM36m in conjunction with the modified TIP3P water model, by performing enhanced sampling MD simulations of several peptides in dilute solutions and conventional MD simulations of globular proteins in dilute and crowded solutions. In our simulations, 10% increase of protein-water LJ interaction for the CHARMM36m cannot maintain stability of a small helical peptide, (AAQAA)_3_ in a dilute solution and only a small modification of protein-water LJ interaction up to the 3% increase (λ = 1.03) is allowed. The modified protein-water interactions are applicable to other peptides and globular proteins in dilute solutions without changing thermodynamic properties from the original CHARMM36m. However, it has a great impact on the diffusive properties of proteins in crowded solutions, avoiding the formation of too sticky protein-protein interactions.

## 1. Introduction

Atomistic descriptions of biomolecules using molecular dynamics (MD) simulation are important to investigate structure-dynamics-function relationships [1,2]. Long-time MD simulations provide us reliable thermodynamic and kinetic properties of proteins and other biomolecules in solution, membrane, and other cellular environments, if accurate molecular force fields are available [3,4]. Atomistic MD simulations used to focus on conformational dynamics of small globular proteins, while large membrane proteins, nucleic acids, and protein-nucleic acid complexes like ribosome have become target systems of the simulations in these days [5,6,7,8]. Intrinsically disordered regions/proteins (IDRs/IDPs) and their assemblies including aggregations and liquid droplets in cells have also been examined using atomistic [9,10,11,12] and coarse-grained (CG) MD simulations [13,14,15,16,17,18]. All of these molecules are necessary for modeling highly heterogeneous and crowded cellular environments [19,20].

To investigate protein structure, dynamics, and function in cellular environments, the effect of macromolecular crowding, or the excluded volume effect, is crucial [21]. In addition, weak and non-specific interactions between macromolecules or between macromolecules and metabolites play important roles, as observed in recent experimental [22,23,24] and computational studies [19,20,25,26,27,28,29]. Such interactions are also used to form liquid-liquid phase separations (LLPSs) in the cytoplasm, membrane, and nucleus, which could be functional platforms for biomolecules [12,30,31]. However, water is the most abundant molecules even in the cellular environments [32]. It suggests that a proper balance between protein-protein and protein-water interactions is necessary for reliable thermodynamic and kinetic properties of proteins predicted in atomistic MD simulations [27,33,34,35].

Molecular force fields for classical MD simulations are usually described as a sum of bonded (bond, angle, and dihedral) and non-bonded (electrostatic and van der Waals) interactions [36,37]. The reliability of the MD simulations strongly depends on the quality of the molecular force field. Therefore, continuous improvements of the molecular force-field parameters have been carried out [38,39,40]. Using the latest well-known force fields like AMBER [41,42], CHARMM [43,44], and OPLS [45], conformational stability and dynamics of globular native proteins are well reproduced in atomistic MD simulations, although there are still debates on protein folding pathways or mechanisms predicted in the simulations [46]. In contrast, IDR/IDP structures and intermolecular interactions between them are difficult to be predicted in MD simulations [9,40,47]. The radius of gyration (*R*_g_) for IDR/IDP predicted in the MD simulations tends to be more compact than that observed in experiments [34,47]. Crowded protein solutions, which mimic the cytoplasmic environments in the cells, are also difficult to be investigated in the MD simulations with conventional molecular force fields [19,20,25,26,27,28,29]. In the simulations, proteins could be aggregated more strongly than experiments, and thereby, translational, and rotational diffusive motions of proteins become slower [27]. To overcome these problems, too sticky protein-protein interactions should be avoided from the conventional molecular force fields.

Since water is the most abundant molecules in most of biological simulations, it is reasonable to modify protein-water interactions, without changing protein-protein interactions from the original ones. Based on the idea, Best and his colleagues suggested the optimal scaling parameter of Lennard-Jones (LJ) interactions between protein and water [34]. With the scaling factor, λ = 1.1 (10% increase of protein-water interaction) for AMBER ff99SB [48] and ff03 [49] in conjunction with TIP3P [50] or TIP4P/2005 water models [51], they successfully simulated structures and dynamics of IDPs including Cold-shock protein and ACTR, consistent with experimental data. As well as in the case of all-atom force fields, the scaling of protein-water interaction in Martini 3 CG model yielded the improved agreement with the small-angle X-ray scattering (SAXS) experiments for IDPs [52]. Recently, Tang et al. further modified the backbone torsion angle parameters of Ser, Thr, and Gln, to improve ff99SBws in conjunction with TIP4P/2005 (ff99SBws-STQ) [35]. Nawrocki et al. also proposed a scaling parameter, λ = 1.09, for CHARMM36 [43] in conjunction with the modified TIP3P model [53] to avoid aggregations of villin headpiece subdomains (villins) in their high-concentration solution [27]. Using the modified force fields, they could investigate slowing down of translational/rotational diffusions of villins, as the concentration of the proteins increases. The slowdown of diffusions was related to the transient cluster formations in crowded solutions, which was also observed by Hummer and coworkers in their extensive µs MD simulations [28] using Amber99SB*-ILDN-Q [54] in conjunction with TIP4P-D water model [55].

These studies suggest the importance of a proper balance between protein-protein and protein-water interactions, which can be controlled via optimizing the scaling parameter, λ, of protein-water LJ interactions. Although similar scaling parameters were found for AMBER ff99SB/ff03 and CHARMM36, the values are not necessarily applicable to other force fields. Our aim, in this study, is to find an optimal scaling parameter for the simulations of globular and disordered proteins in dilute or crowded solutions, when we employ CHARMM36m [44] in conjunction with the modified TIP3P water [53]. We first examine various scaling parameters for LJ interactions between protein and water in MD simulations of (AAQAA)_3_ in solution. To confirm the most reasonable scaling parameter, λ, several proteins including TDP-43 in the C-terminal domain (CTD) [31], chignolin [56], c-Src kinase [57], and villin [58] are simulated in dilute or crowded solution. Considering the slow folding/unfolding processes of these peptides/proteins, enhanced sampling algorithms like replica-exchange molecular dynamics (REMD) [59] for (AAQAA)_3_, Gaussian accelerated MD (GaMD) [60] for TDP-43, and Gaussian accelerated replica-exchange umbrella sampling (GaREUS) [61] for chignolin were employed. We employed different protocols based on the features of each peptide and analysis. In the case of (AAQAA)_3_, the calculation of the temperature dependence of helicity is required for comparison with the experiments. The REMD method is suitable for evaluating the temperature dependence while enforcing the conformational sampling. Since chignolin is a *β*-hairpin forming protein and its timescale for folding/unfolding process is longer than that for *α*-helix, enhanced sampling methods using the collective variables (CVs), such as GaREUS, are needed. TDP-43 is intrinsically disordered, and any CVs are not needed. Thus, GaMD can be used for accelerating the conformational changes without increasing the computational cost from the cMD. Following the work by Nawrocki et al. [27], we test the scaling parameters of protein-water LJ interactions in the range between 1.00 and 1.09. The most reasonable value that we obtain for CHARMM36m in conjunction with the modified TIP3P is largely different from the previously proposed values for AMBER ff99SB/ff03 and CHARMM36, suggesting that a better balance between protein-protein and protein-water interactions is realized in CHARMM36m [44]. However, the small scaling value that we suggest in this study could change the diffusive properties of proteins in crowded solutions.

## 2. Results

### 2.1. Folding and Stability of Small Peptides in Solution

#### 2.1.1. (AAQAA)_3_

We performed REMD simulations [59] of (AAQAA)_3_ using 58 replicas covering the temperature range between 275.0 and 382.0 K. In the simulations, we aim to examine the temperature dependence of the helicity with different scaling parameters, λ, for protein-water LJ interactions. Since the scaling is introduced in CHARMM using the NBFIX function, the scaling parameter, λ, and the term, NBFIX, are used as the same meaning in this paper. Following to the previous studies [44], we regard that the corresponding residues form an *α*-helix, when the backbone dihedral angles (*ϕ* and *ψ*) for three or more consecutive residues satisfy −100° < *ϕ* < −30° and −67° < *ψ <* −7°, respectively. The fraction of helix is computed from the number of residues forming *α*-helix, divided by the total number of residues. In Figure 1a, five curves obtained in the REMD simulations with λ = 1.00 (CHARMM36m), 1.03, 1.04, 1.06, and 1.09, are compared to the experimental results [62]. Although none of the simulation curves fit to the temperature dependence observed in the experiment, the fraction of helix at 300 K for λ = 1.00 (CHARMM36m) (0.17 ± 0.01) and 1.03 (0.15 ± 0.01) are close to the experimental one (~0.2). Note that at λ = 1.00 in our simulation shows similar value to the previously reported in the CHARMM36m paper [44], despite the difference in water box size. In contrast, using larger scaling values, λ = 1.06 and 1.09, the fraction of helix is drastically reduced to less than 0.1 at all the temperatures. Even at 300 K, it is around 0.05, which is much smaller than the experimental value and those with λ = 1.00 and 1.03. Interestingly, the curve with λ = 1.04 shows medium fractions of helicity between λ = 1.03 and 1.06. In Figure 1b, the faction helix at 300 K versus the scaling parameter, λ, is well fitted with the sigmoidal curve,
(1)$$f\left(\lambda \right)=\frac{A}{1+\exp\left(-\left(\frac{\lambda -B}{C}\right)\right)}+D,$$
and the fitted parameters are (*A*,*B*,*C*,*D*) = (0.11, 1.04, −0.0049, 0.0055). Parameter *B* corresponds to the scaling factor for the middle point of the helicity curve.

In Figure 1c, the fraction of helix in each residue at 300 K is compared between the REMD simulations with the scaling parameter, λ, and the experimental results. The residue profiles with λ = 1.00 and 1.03 are similar to the experimental one, although that in residue 3 (Q_3_) is largely underestimated in the simulations. Interestingly, these profiles are also comparable to those predicted in recent AMBER force fields for IDPs (ff99SBws-STQ [35]). In other profiles with λ = 1.04, 1.06, and 1.09, the fraction of helix in each residue is scaled down almost uniformly from those with λ = 1.00 and 1.03.

We next examine the conformational spaces, which were explored in the REMD simulations with different scaling parameters. For this purpose, the two-dimensional potential of mean forces (2D-PMFs) at 300 K are shown along with the end-to-end distance, *d*, and the C*α* root mean square deviation (RMSD) from the ideal α-helix conformation (Figure 2). The folded structures are localized at *d*~24 Å and RMSD~0.5 Å, while the unfolded structures have a broad region with RMSD > 4 Å. Except for λ = 1.09, these two basins (for the folded and unfolded ones) clearly exist in all the PMFs. The PMFs with λ = 1.03 is close to that with the original CHARMM36m (λ = 1.00), while the unfolded basin is slightly emphasized due to stronger protein-water interactions. At λ = 1.04, 1.06, and 1.09, the unfolded basins show deeper and wider free-energy minima as the protein-water interaction increases via the scaling values. These analyses suggest that the shape of the free-energy landscapes of (AAQAA)_3_ does not change significantly with different scaling parameters, λ, while the populations of folded and unfolded states are drastically altered.

The REMD simulation results of (AAQAA)_3_ suggest that the scaling parameter, λ = 1.04, or larger values could underestimate the stability of *α*-helix, when it is applied to CHARMM36m in conjunction with the modified TIP3P. Unexpectedly, the previously proposed scaling parameter for CHARMM36, λ = 1.09, does not work well for (AAQAA)_3_. Since the peptide contains only two types of amino acids (Ala and Gln), more through tests are required to examine the applicability of λ = 1.03 in many other protein systems. Hereafter, we mainly discuss two scaling parameters, λ = 1.00 and 1.03 and examine several protein systems in dilute or crowded solution, whereas the results with λ = 1.09 are presented in Appendix A.

#### 2.1.2. Chignolin

The folding of chignolin (PDB ID: 1UAO [56]), a *β*-hairpin forming protein with 10 amino-acid residues (sequence: GYDPETGTWG), is simulated with GaREUS [61], where replica-exchange umbrella sampling (REUS) [63] and Gaussian accelerated MD (GaMD) [60] are combined. Since a folding process of a *β*-hairpin takes longer time than α-helical peptide, a powerful sampling method, such as GaREUS, is quite useful to get the converged thermodynamics within reasonable computational times. Experimentally, chignolin is reported to form both a stable *β*-hairpin and misfolded structures, where the folded population from the NMR measurement is ~60%. We first test two scaling parameters, λ = 1.00, and 1.03 for the GaREUS simulations of chignolin in water and compare them with λ = 1.09.

Figure 3a shows the probability densities along the C*α*-RMSD with respect to the NMR structure are shown. In the CHARMM36m paper [44], chignolin, and its double mutant, CLN025 were simulated using REMD, showing the native populations of 2.6% and 41%, respectively. Interestingly, the folded population in the simulation with λ = 1.00 (CHARMM36m) has increased to 27%, suggesting the superiority of sampling efficiency in GaREUS compared to REMD for *β*-hairpin peptides. As λ increases, the folded conformations (RMSD < 2.2 Å) increases up to 33% for λ = 1.03. Note that the convergence check of the folded population using different trajectory lengths indicates that folded populations are well converged in the present simulation (Appendix A). The decrease of the folded population is observed for λ = 1.09 (22%) (Appendix A). In comparison, the folded population for λ = 1.09 is lower than those for λ = 1.00 and 1.03. Although the populations for λ = 1.00 and 1.03 are still lower than the NMR measurement [56], the performance of CHARMM36m for the structure prediction of *β*-hairpin peptides is not too bad as discussed in the original paper. Figure 3b shows the 2D-PMFs spanned with the ASP3-GLY7 and ASP3-THR8 distances, *d*(ASP3 − GLY7) and *d*(ASP3 − THR8) which are often used to describe the folded, misfolded, and unfolded states of chignolin [64]. As revealed in the previous study, the regions around (*d*(ASP3 − GLY7), *d*(ASP3 − THR8))~(3 Å, 6 Å) and ~(6.5 Å, 3 Å) correspond to the folded and misfolded states. The overall shapes of the 2D-PMFs for λ = 1.00 and λ = 1.03 are similar to each other. However, the misfolded population for λ = 1.03 is slightly increased and the unfolded population is instead reduced compared to that for λ = 1.00. It is interesting that the folded population of the *β*-hairpin increases as the protein-water LJ interaction is slightly stronger, while the detailed mechanism is still unknown.

#### 2.1.3. TDP-43 in the CTD

Transactive response DNA binding protein 43 (TDP-43) is a versatile nucleic-acid binding protein, playing a central role in amyotrophic lateral sclerosis (ALS) pathogenesis. TDP-43 consists of a well-folded N-terminal domain (NTD), two-highly conserved RNA-recognition motifs (RPMs), and an unstructured prion-like C-terminal domain (CTD). Residues 320–334 in the CTD are found to form a transient *α*-helix in the previous studies with MD simulations and NMR spectroscopy [31]. We therefore select the residues 310–340 of TDP-43 as the second target system. In the simulations, we focus on the secondary structure formation of TDP-43 in the CTD and use the GaMD method [60] as an enhanced sampling method, which can reduce energy barriers between minimum energy states using a GaMD boost potential. We performed ten replicas of GaMD simulations (each for 1 µs) and took the averages of the fraction of helix in each residue. The effect of GaMD boost potential was removed by the reweighting scheme with the cumulant expansion proposed by Miao et al. [60].

The GaMD simulations were conducted with the scaling parameters of λ = 1.00 and λ = 1.03 to make a comparison with the experiment (Figure 4). The fractions of helices show large statistical errors, as the averaged values are just taken from ten independent GaMD runs. Comparison of λ = 1.00 and 1.03 show a small reduction of helicity in the CTD of TDP-43, as protein-water LJ interaction is stronger. For λ = 1.00, the computed helicity in residues 320–330 (region I, sequence: PAMMAAQAA) is in accord with the experimental values, while the helix formed around 335–345 (region II, sequence: GMMGMLASQQ) is found to be too stabilized. The stability of helix of region II is reduced with λ = 1.03, while the reduction of helicity in region I is also observed. In the case of λ = 1.09, the helicity of region II becomes close to the experimental observation, but the difference from the experiment in region I is further emphasized (Appendix A).

The results suggest that the 1.03 scaling of protein-water LJ interaction keeps major structural features of TDP-43 in the CTD, with some differences from the NMR experimental data. Rg of TDP-43 in the CTD with λ = 1.03 does not change from that with λ = 1.00 (Appendix A). It suggests that the scattering function of TDP-43 corresponding to the SAXS profile, which reflects the averaged feature of the protein structures, would be unaltered using the present modification of protein-water LJ interactions. To reduce helicity in the region II, fine tuning of the backbone torsional angles might be necessary like the previous studies of AMBER ff99SBws-STQ [35].

### 2.2. Dynamics and Stability of Globular Proteins in Solution

#### c-Src Kinase

The c-Src kinase (PDB ID: 1Y57) [57] is one of the essential protein kinases, which consists of the kinase domain (276 residues), SH2, and SH3 domains. Although these two domains are necessary for the activation, we simulate only the kinase domain in water as the previous computational works [65,66]. Five independent MD trajectories for each value of λ (1.00 and 1.03) are analyzed to examine the effect of the scaling parameter, λ = 1.03, on globular protein structures (Figure 5). For comparison, we also performed one MD simulation for λ = 1.09, The distribution of the C*α*-RMSDs excluding the residues near the N- and C-terminals have a peak around ~2.5 Å for λ = 1.00 and 1.03 (Figure 5a), suggesting that the kinase domain stability is kept in both cases. Interestingly, the population of the native state (<3.0 Å) with λ = 1.03 slightly increases compared to that with λ = 1.00. The distribution for λ = 1.09 shows a peak spreading around the RMSD of 4.5 Å, which is absent in the cases of λ = 1.00 and 1.03 (Appendix A). It indicates the distortion of the kinase domain structure. Referring to Figure 5b, the C*α* root mean square fluctuation (C*α*-RMSF) analysis shows that the difference is hardly discernible between λ = 1.00 and λ = 1.03, except for the first 40 residues and the residues around 210–220.

The time-series of the C*α*-RMSDs with all the 276 residues in Appendix A reveal larger deviations for λ = 1.03. This suggests that the conformational fluctuations near the N- and C-terminals are enhanced as protein-water LJ interaction increases (with λ = 1.03). The results of the C*α*-RMSDs with and without N- or C-terminal residues (10 residues for each terminal) are reasonable, since we do not change intra-protein interactions of the CHARMM36m, but slightly emphasize protein-water interactions using the scaling parameter of λ = 1.03. The latter seems to affect the motions of N- and C-terminal residues in water, primarily, while the globular domain stability is almost unaltered. Note that the C*α*-RMSD and C*α*-RMSF obtained from the MD simulations with the LJ scaling parameters, λ = 1.00 and 1.03, are similar to those in a previous study using the AMBER ff99SB-ILDN [48]. Since the AMBER force field was tuned to reproduce the conformational dynamics and stability of folded native structures, the results obtained here seem to be promising for simulating the folded native protein structures in solution.

### 2.3. Structural Stability and Diffusivity of Globular Proteins in the Crowded Solutions

Finally, a small globular protein, villin headpiece subdomain (villin), was simulated with the conventional MD simulations in dilute solution as well as in crowded solution. Villin contains 35 amino-acid residues, composing of three *α*-helices [PDB ID: 1VII [58]]. Due to the small size and conformational stability, it was often used to test simulation protocols, folding mechanisms [46], and the effect of macromolecular crowding and weak non-specific interactions [25,26,27]. Here, we prepared two systems: in one system, a single villin is simulated in water (dilute solution) and in the other, eight villins are simulated in solution (crowded solution). Note that in the crowded solution, both target and crowder proteins are villins. The concentration of the crowded solution is about 32 mM, which is the same as those used in our previous study [27].

The time-series of C*α*-RMSDs with respect to the crystal structure in the dilute solution (Appendix A) reveals that villin keeps the native structure (2–3 Å) during the simulations with λ = 1.00. As for λ = 1.03, the RMSDs fluctuate around 2–3 Å for five trajectories, while one trajectory shows larger values of RMSD (~4 Å) as compared to the others due to the orientational change of the N-terminal *α*-helix (Appendix A). On the other hand, the structural characterization with the DSSP algorithm [67,68] reveals that the native secondary structures in the native structure are well conserved for all the trajectories (Figure 6a). In the crowded solution, most villins keep the structures close to the native state (2–3 Å) with λ = 1.00 (Appendix A), while a partial unfolding is observed in one of eight villins. This is rather consistent with our previous studies, which suggests the partial unfolding due to the weak and non-specific interactions. At λ = 1.03, the number of villins showing the large fluctuation compared with the dilute solution increases, while there was no unfolding trajectory in the eight villins (Appendix A. The secondary structures are conserved in the crowded solution (Figure 6b), which suggests that tertiary structures might be broken due to protein-protein interactions. By further increasing the scaling factor up to λ = 1.09, the breakdown of the native structure is observed both for the dilute and crowded solutions (Appendix A).

The translational diffusions of villin in the dilute and crowder solutions are examined. Note that the diffusion coefficient depends on the system size due to the periodic boundary condition (PBC). In the present study, we employ the correction proposed by Yeh and Hummer [69], represented as
(2)$$D=D_{\mathrm{MSD}}+D_{\mathrm{PBC}}\left(R_{h},\eta ,L\right),$$
where *D*_MSD_ and *D*_PBC_ are the diffusion coefficient obtained from the slope of the mean square displacement (MSD), and correction term for PBC defined from the hydrodynamic radius of villin (*R_h_*), solvent viscosity (*η*), and box length of the system (*L*). The value of *R_h_* is taken from HYDROPRO [70] estimation as 13.86 Å. The viscosity of the TIP3P water, *η*_TIP3P_ = 0.35 cP [71], is used for the dilute solution. As for the crowder solution, the Einstein’s relationship between the crowder volume fraction (*ϕ*) and viscosity of the crowder solution (*η_c_*) for suspension is employed. In addition, we utilize the viscosity correction to the diffusion coefficient with the experimental water viscosity, *η*_expt_ = 0.89 cP [72]. The corrected coefficient is defined as *D*′. Further description is available in the Appendix A.

In dilute solution, the values of *D*_MSD_ for λ = 1.00 and 1.03 obtained from the linear fitting of the MSD at 30 ≤ *t*/ns ≤ 50 are also close to each other (Table 1). The corrected coefficients (*D*′) are 0.19 and 0.20 nm^2^/ns for λ = 1.00 and 1.03, respectively. These values after the PBC and viscosity corrections are in excellent agreement with that predicted from HYDROPRO, 0.18 nm^2^/ns. Note that the HYDROPRO predictions are generally close to the experimental values under the dilute condition, and hence this agreement indicates the reliability of CHARMM36m with λ = 1.03 about the description of the diffusive properties as well as the original CHARMM36m for the dilute solutions.

As for the crowder solution, the diffusion coefficients from the MSDs (*D*_MSD_) are 0.031 nm^2^/ns for λ = 1.00 and 0.048 nm^2^/ns for λ = 1.03, respectively (Figure 7b and Table 1). Also, the diffusion coefficient with the PBC correction (*D*_PBC_ is 0.20 nm^2^/ns) is for λ = 1.03, which is close to that from CHARMM36 with λ = 1.09, 0.22 nm^2^/ns [27]. The small modification of protein-water LJ interaction leads to ~1.5 times acceleration of translational motions, judging from MSD results (Figure 7b). These results suggest that the optimal scaling parameter, λ = 1.03, does not change conformational stability of globular proteins significantly, while it avoids from slowdown of translational diffusion for proteins in crowded solution, probably reducing too sticky protein-protein interactions in the MD simulations.

### 2.4. Solvation Free Energies of Amino Acid Analogues

So far, we used the single scaling parameter, λ, to modify protein-water LJ interactions in MD simulations of peptides or proteins. To understand how the modification of the CHARMM36m affects molecular interactions between water and each amino acid, we compute the solvation free energies of the 14 amino acid homologs, associated with the solubility of molecules, through free energy perturbation (FEP) method (Figure 8). For all the species, CHARMM36, CHARMM36m (λ = 1.00), and the modified CHARMM36m (λ = 1.03) are compared to the experimental values of Δ*G_solv_* [73]. Figure 8 shows that all the calculations with λ = 1.03 become closer to the experimental results compared with others, suggesting that the modification seems to be valid for each amino acid. However, to achieve the quantitative agreement with the experiments, in particular, for the analogues of Asn, Gln, Met, Trp, and Tyr, a larger scaling factor is required. For instance, to achieve the quantitative agreement of solvation free-energies between FEP and experiments, about a 40% increase of protein-water LJ interactions (λ = 1.40) is necessary, which may not be applicable to MD simulations of peptides and proteins in solution. To resolve this inconsistency, furthermore careful tunings in molecular force fields are necessary not only for proteins but also for water molecules.

## 3. Discussion and Conclusions

In this study, we performed enhanced sampling MD simulations of small peptides in dilute solution and conventional MD simulations of globular proteins in dilute and crowded solutions using various scaling parameters of protein-water LJ interactions. For CHARMM36m in conjunction with the modified TIP3P water models, about 10% increase of the protein-water interaction affects peptide conformational stability significantly and only small increases (up to 3%) is allowed to keep a good balance between protein-protein and protein-water interactions. The modified force field is applicable not only to *α*-helical but also *β*-hairpin peptides and globular proteins in dilute solution. As enhanced sampling methods, we used REMD for (AAQAA)_3_, GaREUS for chignolin, and GaMD for TDP-43 at the CTD in dilute solution. The GaREUS simulation for chignolin shows higher populations of the folded conformation both for the original (λ = 1.00) and modified (λ = 1.03) CHARMM36m. Note that the original CHARMM36m paper suggested much lower population of the folded state using T-REMD simulation. The GaREUS can enhance conformational sampling with the boost potential, replica-exchange, and umbrella potential for collective variables, which is more suitable to study slow folding/unfolding simulations of a *β*-hairpin peptide than T-REMD. It suggests the importance of enhanced conformational sampling methods for evaluating the quality of molecular force fields for different peptides or proteins.

In crowded villin simulations, translational and rotational diffusive motions are significantly affected by the small modification in the force field, keeping the conformational stability of villin compared to the simulation with the original CHARMM36m force field. This suggests that 3% increase of protein-water LJ interactions reduces too much sticky protein-protein interaction of the original force fields to avoid aggregation in the high-concentration protein solution. It should be noted that the over sticky protein association could lead to the underestimation of the protein diffusivity, and hence the modification of the protein-water interactions proposed by the present study is useful for elucidating the kinetics in the crowded environments.

For IDRs/IDPs simulations, the scaling of protein-water LJ interactions might not be sufficient to explore disordered conformations in dilute and crowded solutions. Although too much sticky protein-protein or intra-protein interactions are avoided, local conformational properties, such as dihedral angle distributions, should be simulated more carefully. As introduced in AMBER ff99SBws-STQ, a careful high-tuning in the backbone dihedral angle potential energy is useful in the simulations of IDRs/IDPs.

The test systems in this study include an *α*-helical peptide, (AAQAA)_3_, a *β*-hairpin peptide, chignolin, a disordered region with some helical fraction, C-terminal TDP-43, globular proteins, c-Src kinase and villin in dilute and crowded solutions. Ideally, a single molecular force field is applicable to such a variety of peptides and proteins in different conditions, predicting their intrinsic structures, dynamics, and interactions at high precisions. This study provides a minimal set for testing the quality of the existing force fields.

## 4. Materials and Methods

All the MD simulations were performed using the GENESIS software [74,75]. CHARMM36m [44] with or without NBFIX corrections are used together with TIP3P water model. After modeling each protein/peptide and adding water molecules and ions, energy minimization was carried out and then short MD simulations were performed for equilibration in NPT and NVT ensembles. The system temperature and pressure were controlled using the Bussi thermostat/barostat [76] with a time step of 2 fs. Water molecules and protein/peptide bonds involving hydrogens were constrained with SETTLE and SHAKE [77], respectively. Long-range electrostatic interaction was calculated using particle mesh Ewald (PME) method [78], while LJ interaction was smoothly reduced to zero from 10–12 Å using a switch function.

### 4.1. (AAQAA)_3_

Initial structure of (AAQAA)_3_ was built as an extended conformation using VMD plugin. The N- and C-terminal residues were capped with the acetyl and amide groups, respectively. The peptide was solvated in a periodic box (63 × 63 × 63 Å^3^), which contained 7826 water molecules. Five NBFIX scaling values for protein-water interactions, λ = 1.00, 1.03, 1.04, 1.06, and 1.09, were tested. To explore a wide conformational space of (AAQAA)_3_, including an *α*-helical, unfold, and extended structures, temperature replica-exchange MD (T-REMD) [59] with 58 replicas were performed. The temperatures for 58 replicas were 275.00, 276.64, 278.29, 279.95, 281.61, 283.29, 284.97, 286.66, 288.36, 290.07, 291.78, 293.50, 295.23, 296.97, 298.72, 300.48, 302.24, 304.02, 305.80, 307.59, 309.39, 311.20, 313.02, 314.84, 316.68, 318.50, 320.36, 322.22, 324.09, 325.97, 327.86, 329.75, 331.66, 333.58, 335.50, 337.44, 339.38, 341.34, 343.30, 345.27, 347.26, 349.24, 351.25, 353.26, 355.28, 357.31, 359.35, 361.40, 363.46, 365.53, 367.61, 369.71, 371.81, 373.92, 376.04, 378.17, 380.00, and 382.0 K, respectively. The temperatures were determined using the web server, REMD Temperature generator [79].

In production runs, each replica was simulated for 525 ns with NVT ensemble. Replica exchanges were attempted every 1500-time step. The equations of motion were integrated using the r-RESPA multiple time step method [80] with a 3.5 fs time step for fast motions and a 7.0 fs time step for slow motions. To make the simulation trajectories stable, optimized hydrogen repartitioning (HMR) and group-based temperature/pressure approach (group T/P) [81] were employed. Structures and energies were written every 1500 steps. Trajectories from 105 to 525 ns were used for analysis.

### 4.2. Chignolin

Initial structure of chignolin was taken from the protein data bank (PDB ID: 1UAO [56]) and was solvated in a periodic box (80 × 80 × 80 Å^3^). The simulation system consists of a chignolin, 2 Na^+^, 17,500 water molecules. Three NBFIX scaling values for protein-water interactions, λ = 1.00, 1.03, and 1.09 were tested. Since folding of chignolin is known as slow processes, we used Gaussian accelerated replica-exchange umbrella sampling (GaREUS) [61] as the enhanced sampling method. In GaREUS, global motions are enhanced with the GaMD biasing potential, while umbrella potentials along the pre-defined collective variables (CVs) were additionally used to enhance conformational sampling along the CVs. In the simulations, we selected the C*α* atom distance between Gly1 and Gly10 as a CV in GaREUS. 24 replicas were simulated, each of which was characterized with harmonic restraint forces along the CV. For λ = 1.00, the force constants and equilibrium distances in the harmonic potentials of the 24 replicas are 0.5, 0.5, 0.5, 0.5, 0.5, 0.5, 0.5, 0.5, 0.5, 0.5, 0.5, 0.5, 1.0, 1.0, 1.0, 1.0, 1.0, 1.0, 1.0, 1.0, 1.0, 1.0, 1.0, 1.0 kcal/mol/Å^2^ and 29.57, 28.32, 26.89, 25.54, 24.43999, 23.15999, 21.87, 20.53999, 19.33999, 18.11999, 16.91999, 15.64999, 14.51999, 13.66999, 12.69999, 11.80999, 10.87999, 9.86999, 8.85999, 8.06, 7.05, 6.35, 5.26, 4.0 Å, respectively. For λ = 1.03 and 1.09, those are 0.5, 0.5, 0.5, 0.5, 0.5, 0.5, 0.5, 0.5, 0.5, 0.5, 0.5, 0.5, 1.0, 1.0, 1.0, 1.0, 1.0, 1.0, 1.0, 1.0, 1.0, 1.0, 1.0, 1.0 kcal/mol/Å^2^ and 29.11998, 27.89998, 26.46998, 25.22998, 23.97998, 22.66998, 21.45998, 20.14998, 18.89998, 17.70998, 16.40998, 15.20999, 14.21999, 13.26999, 12.48999, 11.63999, 10.73999, 9.75999, 8.80, 7.89, 7.05, 6.14, 5.18, 4.0 Å, respectively. The boost potentials in GaMD for λ = 1.00 and 1.03 were initially guessed from an initial 4-ns NVT simulation without boosting. After the initial determination, the boost potentials were updated every 100 ps in a 5-ns NVT simulation while boosting the system. The boost potentials obtained from λ = 1.03 were also used for λ = 1.09. The thresholds in the boosting potentials are set to the lower bound. *σ*_0_ was set to 6 kcal/mol for both potentials.

In production runs, we performed 1-μs GaREUS simulations with NVT ensemble in each replica for λ = 1.00 and 1.03, and 0.425-μs GaREUS simulation for λ = 1.09. To obtain unbiased free-energy landscapes, the two-step reweighting scheme, which consists of the second-order cumulant expansion for removing the GaMD biases and multistate Bennett acceptance ratio (MBAR) [82] for harmonic restraint potentials in REUS, were applied to the simulation trajectories. A 5-fs time step was used to integrate the equation of motion using velocity Verlet integrator with HMR and group T/P. Structures and energies were output every 1000 steps and every 500 steps, respectively. The first 0.1-μs in the GaREUS simulation for each condition were omitted as equilibration.

### 4.3. TDP-43

The C-terminal domain of TDP-43 (residues 310-340) in water was simulated using dual-boost GaMD [60]. The N- and C-terminal residues were capped with the acetyl and amide groups, respectively. The peptide was solvated in a periodic box (65 × 65 × 65 Å^3^), which contain 8395 water molecules, 24 Na^+^, and 24 Cl^-^. Three NBFIX scaling values, λ = 1.00, 1.03, and 1.09, were tested. Initial structure of TDP-43 was first built as an extended conformation. The structure was collapsed using the short MD simulation in vacuum. In dual-boost GaMD, one boost potential is applied to the dihedral and CMAP energy terms and another to the total potential excluding the dihedral and CMAP energies. To obtain the initial guesses of the both boosting potentials, the thresholds and force constants were calculated from an initial 5-ns simulation without boosting. After the initial determination of the GaMD parameters, the boost potentials were added to the system while continuing the update of parameters every 100 ps in a 15-ns NVT simulation. The finally determined parameters were used for production runs. The thresholds in the boosting potentials are set to the lower bound. *σ*_0_ was set to 6 kcal/mol for both potentials.

In production runs, ten independent GaMD simulations for 1 μs each were performed with NVT ensemble, and the results were averaged. A 5-fs time step was used to integrate the equation of motion using velocity Verlet integrator with HMR and group T/P. To analyze the GaMD simulation trajectories, we avoid the effect of GaMD biasing potential using the reweighting scheme based on the cumulant expansions [60]. Structures and energies were output every 20,000 steps and every 2000 steps, respectively. Trajectories from 0.1 to 1 μs were used for analysis.

### 4.4. c-Src Kinase

Initial structure of c-Src kinase was taken from PDB (PDB ID: 1Y57 [57]) and was solvated in a periodic box (102 × 102 × 102 Å^3^). The system consists of 103,592 atoms, where c-Src kinase, 94 Na^+^, 89 Cl^−^, and 33,000 water molecules exist. Three NBFIX values (1.00, 1.03 and 1.09) were used. Five independent conventional MD (cMD) simulations for 1 μs for λ = 1.00 and 1.03 were performed with NVT ensemble, while one simulation for 1 μs was done for λ = 1.09. The r-RESPA integrator with a 3.5 fs were employed with HMR and group T/P. Structures and energies were written every 6000 steps and every 3000 steps, respectively. Trajectories from 0.2 to 1 μs were used for analysis.

### 4.5. Villin Crowding Solution

For the dilute system, one villin was solvated in a periodic box (102 × 102 × 102 Å^3^). The system consists of 99,776 atoms, where villin, 89 Na^+^, 91 Cl^−^, and 33,000 water molecules exist. Three NBFIX values (1.00, 1.03 and 1.09) were used. Five independent cMD simulations for 1 μs each were performed with NVT ensemble for λ = 1.00 and 1.03. As for λ = 1.09, one cMD simulation for 1 μs was performed. The r-RESPA integrator with a 3.5 fs were employed with HMR and group T/P. Structures and energies were output every 6000 steps and every 3000 steps, respectively. The first 0.2 μs trajectory for each was omitted as equilibration.

For the crowded solution, eight villins were solvated in a periodic box (78 × 78 × 78 Å^3^). Initial structure of villin was taken from PDB (PDB ID: 1VII [58]). The system consists of 42,262 atoms, where 8 villins, 34 Na^+^, 50 Cl^−^, 12,466 water molecules exist. Three NBFIX values (1.00, 1.03, and 1.09) were used. The trajectory lengths of cMD simulations with NVT ensemble are 1.5 μs for λ = 1.00 and 1.03, and 1.0 μs for λ = 1.09. The r-RESPA integrator with a 3.5 fs were employed with HMR and group T/P. Structures and energies were output every 15,000 steps and every 5000 steps, respectively. Trajectories from 0.2 to 1.5 μs were used for analysis.

### 4.6. Amino-Acid Analogues

The 14 species of amino-acid side-chain analogues (Ala, Asn, Cys, Gln, Hsd, Ile, Leu, Met, Phe, Ser, Thr, Trp, Tyr, and Val) were used for calculations of absolute solvation free energies. The initial structure of each analogue was taken from the ChemSpider [83] and PubChem databases. Force field parameters for the beta carbon were changed and one hydrogen atom was added to the beta carbon [84]. 14 NBFIX values (λ = 1.00, 1.01, 1.02, 1.03, 1.04, 1.05, 1.06, 1.07, 1.08, 1.09, 1.1, 1.2, 1.3, and 1.4) were used. Each analogue was solvated within a periodic box (50 × 50 × 50 Å^3^), which contains ~4000 water molecules.

For each analogue, the absolute solvation free energy was estimated as Δ*G_solv_* = Δ*G^vacuum^* − Δ*G^water^*, where Δ*G^water^* and Δ*G^vacuum^* are the free-energy changes upon annihilating nonbonded interactions of the analogue in water and in vacuum, respectively. Δ*G^water^* and Δ*G^vacuum^* were calculated using FEP/*λ*-REMD. In FEP/*λ*-REMD calculations, the electrostatic and Lennard-Jones (LJ) interactions of the analogue are scaled by *λ_elec_* and *λ_LJ_*, respectively. The following 24 windows were used with different coupling parameters: *λ_elec_* = 1, 0.9, 0.8, 0.7, 0.6, 0.5, 0.4, 0.3, 0.2, 0.1, 0, 0, 0, 0, 0, 0, 0, 0, 0, 0, 0, 0, 0, 0, and *λ_LJ_* = 1, 1, 1, 1, 1, 1, 1, 1, 1, 1, 1, 0.9, 0.8, 0.7, 0.6, 0.5, 0.4, 0.3, 0.2, 0.15, 0.1, 0.05, 0.025, 0. In each calculation, the simulation was run for 3 ns per each window with NPT ensemble, and trajectories from 1 to 3 ns were used for analysis. Replica exchanges were attempted every 800 steps. The free-energy changes are estimated using the Bennett’s acceptance ratio (BAR) method. The obtained trajectories are decomposed into three blocks. The mean and the standard error were calculated using the block averages. For the water system, the r-RESPA integrator with a 2.5 fs were employed. For the vacuum system, the velocity Verlet integrator with a 2.0-fs time step were employed, and instead of using PME, non-bonded interactions were truncated at a cutoff distance of 1000 Å.

## Figures and Tables

**Figure 1 molecules-27-05726-f001:**
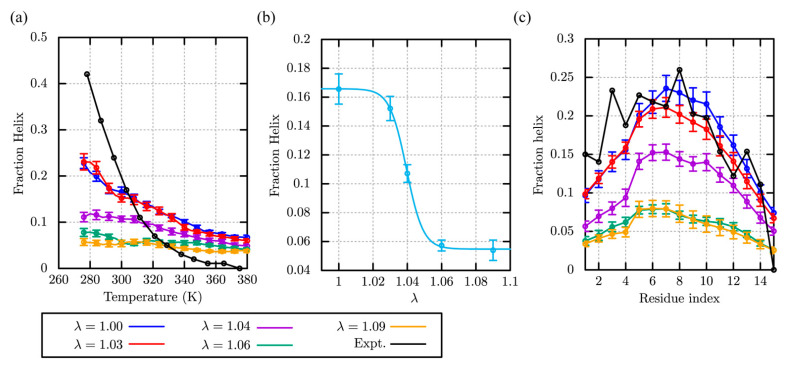
The *α*-helicity of (AAQAA)_3_ using a different scaling parameter, λ. (**a**) Temperature dependency of the fraction of helix, (**b**) the λ-dependency of the fraction of helix at 300 K, (**c**) the fraction of *α*-helix in each residue at 300 K.

**Figure 2 molecules-27-05726-f002:**
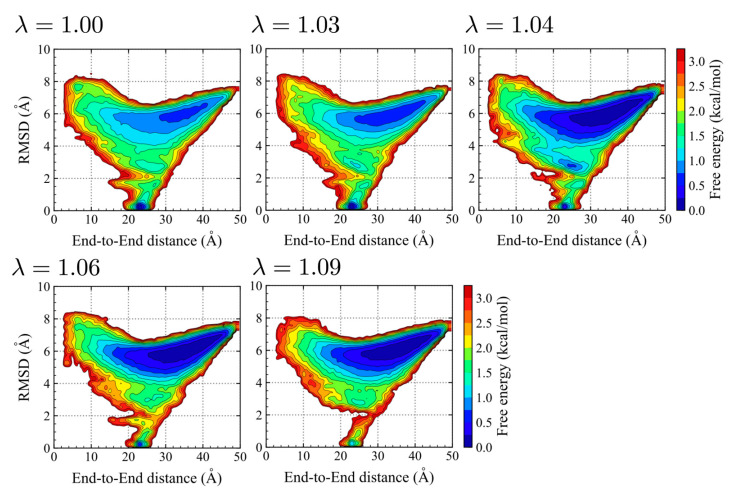
Two-dimensional potential of mean forces (2D-PMFs) of (AAQAA)_3_ on the end-to-end distance, *d*, and the C*α* root mean square deviation (RMSD) from the ideal helix for different λ = 1.00, 1.03, 1.04, 1.06, and 1.09.

**Figure 3 molecules-27-05726-f003:**
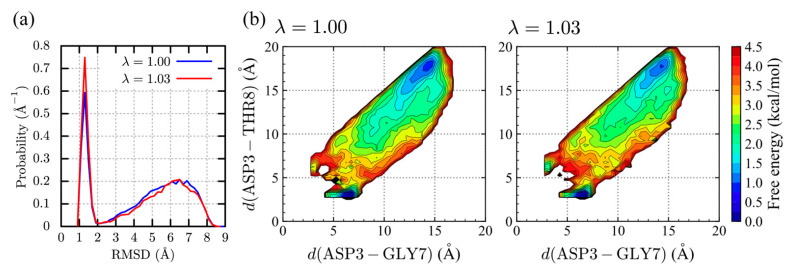
Distributions for the chignolin conformation. (**a**) Probability densities along the C*α*-RMSD. (**b**) 2D-PMFs spanned with the ASP3-GLY7 and ASP3-THR8 distances for λ = 1.00 (**left**) and 1.03 (**right**).

**Figure 4 molecules-27-05726-f004:**
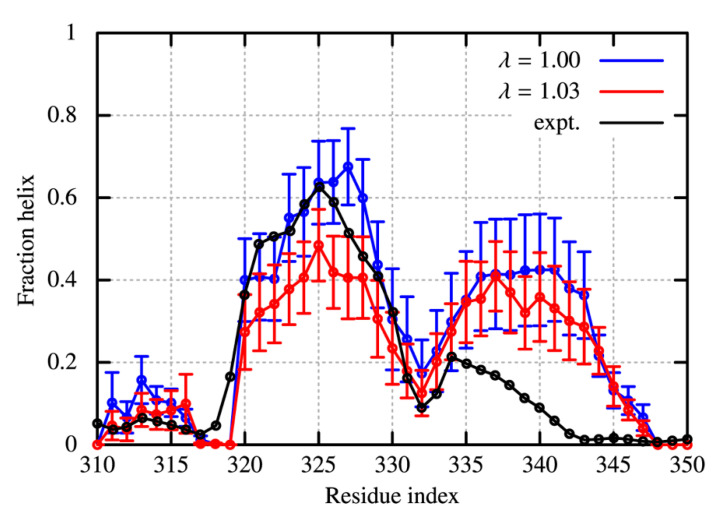
Fraction of helix in the C-terminal domain of TDP-43 for the conditions of λ = 1.00 (blue) and of λ = 1.03 (red), and for the NMR measurement (black). The helicity is evaluated through the DSSP algorithm.

**Figure 5 molecules-27-05726-f005:**
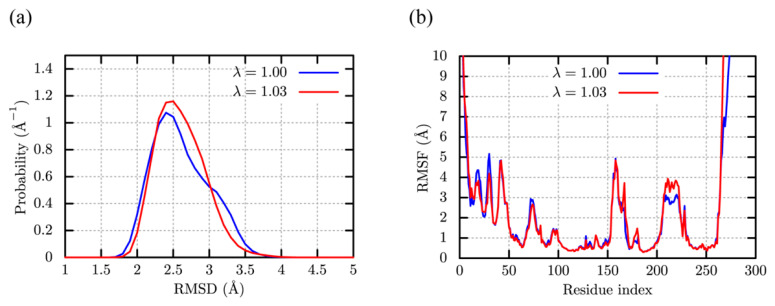
Structural fluctuation of c-Src kinase in water. (**a**) Distribution of C*α*-RMSD. The terminal residues are excluded for the analysis. (**b**) The C*α* root mean square fluctuation (C*α*-RMSF).

**Figure 6 molecules-27-05726-f006:**
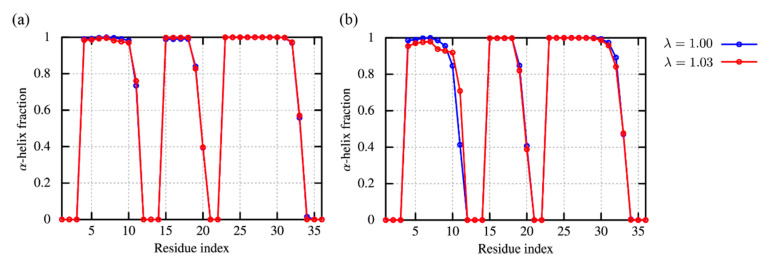
The α-helix fraction of villin in the dilute (**a**) and crowded solutions (**b**). The fraction is computed through DSSP algorithm. As for the crowded solution, the averages of the fraction over eight villins are shown.

**Figure 7 molecules-27-05726-f007:**
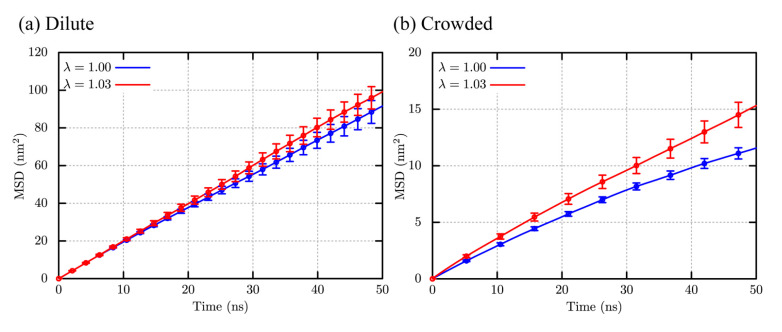
Mean square displacements (MSDs) of villin in (**a**) the dilute and (**b**) crowder solutions.

**Figure 8 molecules-27-05726-f008:**
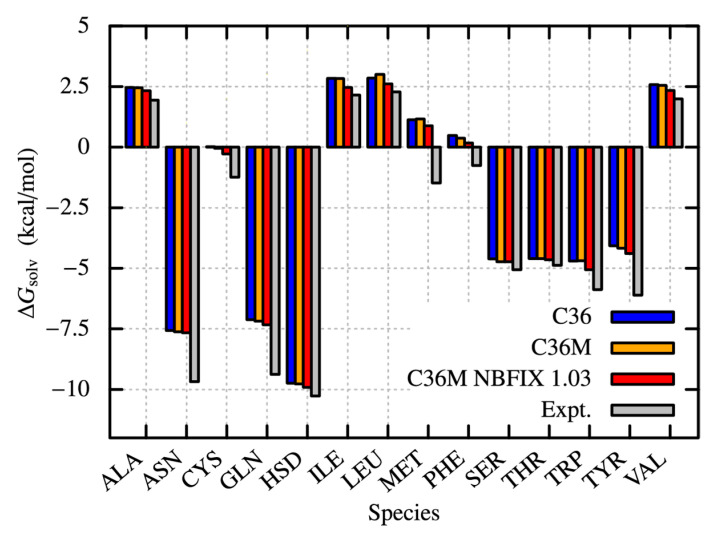
Solvation free energies (Δ*G_solv_*) of amino acid analogues computed using free energy perturbation (FEP) method.

**Table 1 molecules-27-05726-t001:** Diffusion coefficients of villin in the dilute and crowder solutions with NBFIX 1.03 scaling factor. The values for 1.00 scaling are shown in parenesis.

	*D*_MSD_ (nm^2^/ns)	*D*_PBC_ (nm^2^/ns)	*D* (nm^2^/ns)	*D′* (nm^2^/ns)
Dilute	0.33 (0.30)	0.17 (0.17)	0.50 (0.48)	0.20 (0.19)
Crowded	0.048 (0.031)	0.15 (0.15)	0.20 (0.18)	0.077 (0.070)

## Data Availability

The data that support the findings of this study are available from the corresponding author upon reasonable request.

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
