# Peer review of "Modified Protein-Water Interactions in CHARMM36m for Thermodynamics and Kinetics of Proteins in Dilute and Crowded Solutions"

_molecules, 2022, doi:10.3390/molecules27175726_

Round 1

Reviewer 1 Report

This manuscript describes an extensive work to further optimize the broadly utilized CHARMM36m force field to better describe the structure, dynamics and stability of proteins in both dilute and crowded solutions. This work adds to extensive efforts for tuning of general-purposed empirical force fields for protein simulations. It is very timely and expected to be highly impactful. Briefly, this work focuses on re-tuning the protein-water Lennard-Jones interactions using a set of disordered and folded proteins. The key finding is that a scaling of \lamda = 1.03 is more appropriate compared to the original scaling of 1.09 proposed for addressing the over compaction bias of CHARMM36m. This is an important recommendation and will be of great interest to the community. I strongly recommend this work for publication in Molecules. Nonetheless, I have a few minor and modest recommendations that the authors may consider to further improve this strong work.

1. It is curious that three sampling protocols were used for three proteins. It will be very helpful to provide a rationale of this choice, particularly on the pros and cons of each protocol and why each protocol is particularly suitable for each peptide.

2. Convergence should be also provided for chignon results (e.g., Fig 3a). It is a bit puzzling that \lamda = 1.03 yields a larger folded population, as one would expect increasing protein-water interactions should stabilize the unfolded state more.

3. In discussion of stability of villain headpiece in dilute and concentrated conditions, no main figure is provided. Please consider include at least one main figure.

There is a discussion of tertiary stability vs secondary structure stability on Page 8. This discussion can be substantiated by calculating the fraction of tertiary native contacts formed.

4. The analysis of diffusion coefficients on Page 9 is a bit confusing. It is not clear whether \lamda = 1.03 improves these properties or not and what is the reference (experimental?) for drawing such a conclusion. Along this line, it is not clear the last statement on Page 9 is supported by clear data/comparison.

5. This is more work, but I think it makes sense to include the results from \lamda = 1.09 (default in CHARMM36mw) for all systems.

6. Overall, it appears that \lamda = 1.03 has only modest impacts on all properties examined (except helicity of aaqaa). This should be carefully discussed to make a more clear and more convincing case for users to follow the new recommendation.

Reviewer 2 Report

In this paper, the authors have performed molecular dynamics simulation (MD) and examined the effect of modified water-protein interaction on the stability of the helical structure of (AAQAA)3, chignolin, TDP-43, and c-SRC kinase. Here authors use replica exchange molecular dynamics for (AAQAA)3, Gaussian accelerated replica-exchange umbrella sampling for chignolin, and Gaussian accelerated MD for TDP-43. The effect of crowding on the aggregation propensity of villin is also studied. All the simulations were performed to find an optimum value of water-protein interaction scaling parameter (lambda).

This paper highlights the importance of choosing the appropriate force field for simulations. The report concludes that the lambda=1.03 is an appropriate scaling parameter to obtain the right balance of inter- and intra-chain interaction among folded and disordered proteins. The article is well written and deserves publication. However, The Authors need to address these minor points before publication:

  1. A recent paper showed an increase in the water-protein interaction by 10% in the Martini force field and improved the agreement with experiments for disordered protein. This result should be included in the discussion. (J. Chem. Theory Comput. 2022, 18, 2033−2041)

  2. Authors should include a comment on matching the SAXS profile of IDP on changing the water-IDP interaction.

  3. Recent papers on intrinsically disordered proteins such as J. Phys. Chem. Lett. 2021, 12, 37, 9026–9032, and Protein Sci. 2021;30:1371-1379 should be referenced.

Reviewer 3 Report

The authors provide a good introduction to the topic, describing the importance of the research area and highlighting the need for further research on the subject.

The methodology is well described, which in academic terms would enable the development of similar research by other researchers.

Finally, the results and their analysis are significant, the results present high scientific quality. 

In this order of ideas, I recommend the publication of the manuscript without additional modifications.

Author Response

This reviewer did not request any changes from the original manuscript.